# Energy Balance, CO₂ Balance, and Meteorological Aspects of Desertification Hotspots in Northeast Brazil

**Any Caroline Silva [1,*], Keila Rêgo Mendes [1], Cláudio Moisés Santos e Silva [1], Daniele Torres Rodrigues [2], Gabriel Brito Costa [1,3], Duany Thainara Corrêa da Silva [4], Pedro Rodrigues Mutti [1], Rosaria Rodrigues Ferreira [1] and Bergson Guedes Bezerra [1]**

[1] Department of Atmospheric and Climate Sciences, Federal University of Rio Grande do Norte, Natal 59078-970, RN, Brazil; keila.inpa@gmail.com (K.R.M.); claudiom8@gmail.com (C.M.S.eS.); gabrielbritocosta@gmail.com (G.B.C.); pedromutti@gmail.com (P.R.M.); rosa.meteoro.ferreira@gmail.com (R.R.F.); bergson.bezerra@gmail.com (B.G.B.)

[2] Department of Statistic, Federal University of Piauí, Teresina 64049-550, PI, Brazil; mspdany@gmail.com

[3] Institute of Biodiversity and Forests, Federal University of Western Pará, Santarém 68035-110, PA, Brazil

[4] Institute of Engineering and Geosciences, Federal University of Western Pará, Santarém 68035-110, PA, Brazil; duanythaynara@gmail.com

\* Correspondence: anycarolinen@hotmail.com

**Abstract:** The main objective of this study was to evaluate meteorological variables and the simulated components of energy and CO₂ balances in desertification hotspots in Northeast Brazil. Meteorological data were obtained from the National Institute of Meteorology measurement network for the Cabrobó and Ibimirim sites. Initially, hourly linear trends were calculated for the meteorological variables using the non-parametric Mann–Kendall test. Then, the seasonal variability in the components of energy and CO₂ balances was assessed through simulations of the simple tropical ecosystem (SITE) model. Results showed evidence of increasing air temperature trends in the Cabrobó site in the first months of the year, which was not observed in the Ibimirim site. Regarding relative humidity, increasing trends were observed in a few months over the Cabrobó site, while decreasing trends were observed in the Ibimirim site. Opposite behaviors were also identified for the trends in wind speed in both sites. Gross primary production (GPP) and net ecosystem exchange (NEE) simulated values were higher in the first half of the year in both sites. GPP varied from 0.8 to 1.2 g C m⁻² h⁻¹, and NEE fluctuated around approximately −5 g C m⁻² h⁻¹. These results indicate that rainfall seasonality is a crucial factor for the modulation of CO₂ and energy balance fluxes in the Caatinga biome.

**Keywords:** microclimate; land degradation; atmosphere–biosphere interaction; modeling

## 1. Introduction

Desertification can be described as the persistent degradation of the biophysical environment of drylands, caused either by human-induced processes or climate variability [1–3]. The United Nations Convention to Combat Desertification [1] acknowledges it as one of the major environmental problems worldwide, and the associated impacts are estimated to affect up to 250 million people [1,4]. Desertification poses an even bigger threat to already vulnerable arid and semiarid regions, where the local population is usually highly dependent on natural resources, and therefore, land degradation furthers hampers development [5–7]. Furthermore, future climate change scenarios indicate an overall increase in the frequency and intensity of climate extreme events such as droughts, which may accelerate desertification processes over drylands [8].

In South America, the Northeast Brazil (NEB) region encompasses some of the most vulnerable semiarid regions in the continent [9]. Indeed, studies in the NEB already indicate a reduction in rainfall volumes and an expansion of its arid and semiarid domains

[10–12]. Additionally, degraded lands in the NEB are expanding due to increasing fire-wood production and poor soil management and agriculture practices (slash and burn), which cause deforestation, soil salinization, and ultimately desertification [4,13].

The rapid intensification of desertification in the NEB is responsible for the suppression of the Caatinga vegetation, a highly biodiverse, exclusively Brazilian seasonally dry tropical forest, characterized mainly by thorny deciduous species [14]. The degradation of the Caatinga biome may play a fundamental role in the formation and expansion of desertification hotspots in the NEB, as previously reported by others [4,13,15]. Therefore, it is of extreme significance to study and quantify the impacts of desertification in key environmental and biophysical processes related to the native Caatinga vegetation, such as energy and $CO_2$ balances.

For instance, despite the systematic increase in $CO_2$ concentrations in the atmosphere throughout the years, terrestrial vegetated ecosystems usually behave as carbon sinks, and thus, biome-oriented studies are of the utmost importance to better understand $CO_2$ dynamics at the regional and global scales [16,17]. However, the interannual variability in $CO_2$ absorption by terrestrial carbon sinks is strongly related to rainfall, temperature, and other meteorological variables [18,19]. Recent studies showed that the Caatinga biome potentially behaves similar to a carbon sink even in dry years, but $CO_2$ dynamics were largely driven by meteorological conditions [20,21]. The potential impacts of desertification in the $CO_2$ balance of Caatinga are yet unknown.

Similarly, the energy balance analysis over vegetated surfaces allows the dimensioning of mass and energy exchanges in the soil–vegetation–atmosphere system through the partitioning of net radiation into energy processes such as latent (associated with evaporation and transpiration processes), sensible (used to heat the air), and soil (which is associated with heat diffusion in the soil) heat fluxes [22–24]. They also allow the assessment of microclimate changes over a given region due to land-use changes, as well as soil and atmosphere conditions. Therefore, landscape changes due to desertification will impact the partitioning of energy balance [25]. For example, previous studies in the Caatinga biome indicated that 50% of annual net radiation was used as sensible heat flux [24]. The expansion of desertification hotspots can further increase the portion of radiation transformed into sensible heat flux, which may impact the local and regional climate [25].

Currently, there are no studies in the scientific literature with in situ measurements of the components of these balances over desertification hotspots in the NEB, despite the undeniable environmental and social relevance of this process to the region. Thus, the objectives of the present study were (1) to evaluate the role of the different components of energy and $CO_2$ balances in desertification hotspots in the semiarid NEB and (2) to assess the regional dynamics of the desertification process in relation to meteorological variables measured in situ. The energy balance components (latent and sensible heat fluxes), the Bowen ratio (which is the ratio between sensible and heat fluxes), and the $CO_2$ balance components (gross primary production—GPP, and net ecosystem exchange—NEE) were simulated using SVAT models. To that end, we used data measured in the Cabrobó site (NEB desertification hotspot) and in the Ibimirim site (surroundings of the Cabrobó desertification hotspot) while also identifying trends in the meteorological variables measured over the two sites.

The state-of-the-art method for the evaluation of $CO_2$ and energy balances over vegetated environments is the use of eddy-covariance systems through high-frequency continuous measurements [26–28]. However, these systems are costly, and their installation and monitoring in desertification zones may be challenging. Therefore, biophysical soil–vegetation–atmosphere transfer (SVAT) models can be used as alternative tools to simulate the components of $CO_2$ and energy balance. It must be noted that both observational and modeling studies have been carried out in Brazil, mostly over the Amazon and the Cerrado biome [29]. The Caatinga biome and the NEB semiarid region, however, are still scarcely studied regarding $CO_2$ and energy balance, despite recent efforts [21,24,30–32].

## 2. Materials and Methods

### 2.1. Data

Meteorological data were obtained through the National Institute of Meteorology (Instituto Nacional de Meteorologia—INMET)) measuring network. The spatial distribution of the selected stations, as well as the location of the desertification hotspots in the NEB (as delimited by the National Institute for the Semiarid (INSA)) are presented in Figure 1. Air temperature (°C), relative humidity (RH%), atmospheric pressure (hPa), wind speed (m s⁻¹), global radiation (kJ m⁻²), and rainfall (mm) data were used. They were measured every five seconds and stored as hourly means. The exception is rainfall data, which refers to hourly accumulated values. Data were available for the period from 1 January 2008 until 3 December 2018.

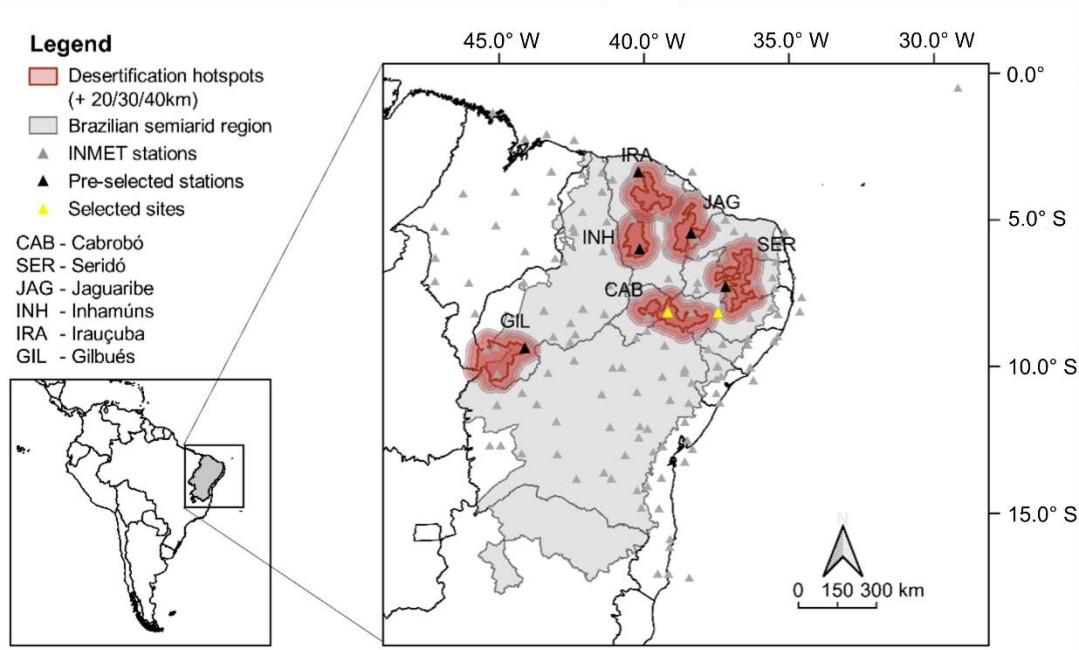

**Figure 1.** Spatial distribution of INMET stations and location of desertification hotspots.

### 2.2. Selection of the Simulation Sites

Firstly, we analyzed seven (which are the black and yellow triangles) cities located within the six NEB desertification hotspots (Figure 1). We included Ibimirim town in the analysis due to the proximity to the CAB desertification hotspots. From the Gilbués hotspot, we selected the Bom Jesus do Piauí town, located at the southern Piauí state, in a transition region between the Caatinga and Cerrado biomes. In the Irauçuba hotspot, the Sobral city was selected, located 20 km from the hotspot boundaries. The Patos town was selected for the Seridó hotspot, which is also 20 km in distance from its boundaries. For the Jaguaribe, Inhamuns, and Cabrobó desertification hotspots, the towns of Jaguaribe, Tauá, and Cabrobó were selected, which are located inside the hotspots. The coordinates, elevation, and percentage of data gaps for each selected station are presented in Table 1. Mean annual accumulated rainfall, temperature, and relative humidity are shown in Table 2. Cabrobó and Ibimirim feature the lowest mean annual precipitation values, with 404.6 and 380.9 mm/year, respectively. However, air temperature is lower in these two locations due to nighttime cooling, which is a typical phenomenon in arid regions, particularly during the drier periods of the year.

These seven stations were primarily analyzed regarding (i) data quality, prioritizing stations with fewer data gaps; (ii) proximity to the desertification hotspot, prioritizing stations located within hotspots; (iii) proximity of other nearby stations in the surroundings of the hotspots that could be used as comparison regarding the response of each observed

and simulated data. By adopting these criteria, the Cabrobó site was selected for the simulations and meteorological data analysis, with the Ibimirim station being used for comparison (Table 1). Both sites are located in the Pernambuco state; the Cabrobó site is located within the Cabrobó desertification hotspot (Figure 1), and the Ibimirim site is located approximately 40 km from the boundaries of the hotspot and 190 km from the Cabrobó site. The selected hotspot and sites are located south of the Pernambuco state, in a region highly dependent on agriculture, which can be directly affected by the consequences of desertification, economically impacting the entire region [33].

**Table 1.** Location of the seven municipalities located within or in the surroundings of each desertification hotspot in the NEB, as shown in Figure 1: PI, Piauí state; PE, Pernambuco state; CE, Ceará state; PB, Paraíba state.

| Station | Latitude | Longitude | Elevation | Gaps |
|---|---|---|---|---|
| Bom Jesus do Piauí (PI) | 09°04′28″ S | 44°21′31″ W | 277 m | 7% |
| Cabrobó (PE) | 08°30′51″ S | 39°18′36″ W | 325 m | 3% |
| Ibimirim (PE) | 08°32′26″ S | 37°41′25″ W | 395 m | 2% |
| Jaguaribe (CE) | 05°53′26″ S | 38°37′19″ W | 123 m | 3% |
| Patos (PB) | 07°01′28″ S | 37°16′48″ W | 242 m | 7% |
| Sobral (CE) | 03°41′10″ S | 40°20′59″ W | 69 m | 15% |
| Tauá (CE) | 06°00′11″ S | 40°17′34″ W | 402 m | 3% |

**Table 2.** Averages of climate variables of the seven municipalities according to Table 1.

| Averages of Climate Variables | | | |
|---|---|---|---|
| Station | Annual Accumulated Precipitation (mm) | Air Temperature (°C) | Relative Humidity (%) |
| Bom Jesus do Piauí (PI) | 598.1 | 27.3 | 62.3 |
| Cabrobó (PE) | 407.6 | 27.1 | 59.1 |
| Ibimirim (PE) | 380.9 | 25.9 | 59.1 |
| Jaguaribe (CE) | 407.8 | 29.1 | 55.0 |
| Patos (PB) | 654.6 | 27.9 | 52.0 |
| Sobral (CE) | 505.1 | 27.6 | 69.0 |
| Tauá (CE) | 507.5 | 27.1 | 55.9 |

*2.3. Soil–Vegetation–Atmosphere Transfer Model*

In order to determine the components of energy and $CO_2$ balances, we used the simple tropical ecosystem (SITE) model, which represents soil–vegetation–atmosphere processes and was first designed as a tool to be used in tropical ecosystems [34]. The SITE model comprises different surface processes such as infrared radiation budget above the canopy, solar radiation budget, aerodynamic processes, canopy physiology and transpiration, canopy water budget, mass and energy transfer in the atmosphere, two-layer soil heat fluxes, soil water, carbon flux, and balance.

There is a great variety of available models similar to SITE [29], as reported in another study. However, for this particular study, the SITE model is more advantageous because it has been previously calibrated for the Caatinga environment [31], presenting satisfactory results, with correlation coefficients statistically significant at a 95% confidence level, which is higher than 0.70 for energy and $CO_2$ balance components. These results were obtained through the assimilation of in situ measurements of biome information such as specific leaf area, leaf width, maximum capacity of rubisco enzyme, coefficient of stomatal conductance, etc. Thus, the same specific parametrization used in another study [31] was used in the simulations of this study.

In order to more objectively represent results, we selected two contrasting years regarding annual rainfall volumes, since it is the main meteorological aspect driving ecosystem activity in the region. The choice of the years was based on another study [2],

which evaluated vegetation–rainfall relations in NEB desertification hotspots and showed that 2009 was an anomalously wet year and 2014 was an anomalously dry year in the Cabrobó hotspot. Table 3 shows the annual accumulated rainfall and its anomalies during the period from 2008 to 2018. The anomalies were calculated by subtracting each annual value from the 2008–2018 average. We can observe that in 2009 the anomalies were most intense in both Cabrobó and Ibimirim, but during 2014, the anomalies were lower than in 2012 or 2017. It is worth mentioning that the 2014 dataset presented fewer gaps than the other years, and an ongoing intense drought episode was established in 2012. Therefore, each simulation started at 00:00 UTC on 1 January of each respective year and ended at 23:00 UTC on 31 December of each selected year.

**Table 3.** Annual accumulated rainfall and annual anomalies to the period from 2008 to 2018 in Cabrobó and Ibimirim.

| | Annual Rainfall Accumulated (mm) | | Annual Rainfall Anomalies (mm) | |
|---|---|---|---|---|
| Years | Cabrobó | Ibimirim | Cabrobó | Ibimirim |
| 2009 | 747.5 | 724.4 | 367.9 | 311.2 |
| 2010 | 502.0 | 835.6 | 122.4 | 422.4 |
| 2011 | 562.2 | 560.8 | 182.6 | 147.6 |
| 2012 | 207.6 | 147.0 | −172.0 | −266.2 |
| 2013 | 321.2 | 338.2 | −58.4 | −75.0 |
| 2014 | 229.4 | 307.8 | −150.2 | −105.4 |
| 2015 | 241.4 | 285.4 | −138.2 | −127.8 |
| 2016 | 386.2 | 327.8 | 6.6 | −85.4 |
| 2017 | 224.4 | 264.0 | −155.2 | −149.2 |
| 2018 | 373.8 | 341.4 | −5.8 | −71.8 |

The SITE model uses six input meteorological variables (hourly samples) as initial conditions: rainfall (mm), air temperature (°C), specific humidity (g kg$^{-1}$), wind speed (m s$^{-1}$), global radiation (W m$^{-2}$), and longwave radiation emitted by the atmosphere (W m$^{-2}$). Output variables simulated by the model are net radiation ($R_n$) photosynthetically active radiation (PAR), soil heat flux (G), sensible heat flux (H), latent heat flux (LE), net $CO_2$ exchange (NEE), net primary production (NPP), gross primary production (GPP), the fraction of absorbed photosynthetically active radiation (FAPAR), and leaf area index (LAI).

Rainfall, air temperature, atmospheric pressure, wind speed, and global radiation are directly measured at INMET stations. Specific humidity ($q$) and longwave radiation emitted by the atmosphere ($L\downarrow$ were calculated based on classic methods that will be described in the following sections.

2.3.1. Specific Air Humidity Calculation

Specific air humidity is defined as the mass of water vapor in a unit of air mass (kg kg$^{-1}$ or g kg$^{-1}$) and can be calculated by using RH, air temperature, and atmospheric pressure data [35]. RH is the ratio between partial water vapor pressure ($e$) and water vapor saturation pressure ($e_s$). Thus,

$$RH = \frac{e}{e_s} \times 100 \tag{1}$$

where $e_s$ is determined by the Tetens equation as follows:

$$e_s = 6.1078 \times 10^{\frac{7,5\,T}{237.3+T}} \tag{2}$$

Where $T$ (°C) is air temperature. Then, we calculate $e$ as follows:

$$e = \frac{RH \times e_s}{100} \tag{3}$$

and finally, specific humidity is determined by the following equation:

$$q = \frac{0.622\,e}{P - 0.378\,e} \tag{4}$$

where $P$ (hPa) is the atmospheric pressure directly measured at the INMET station.

### 2.3.2. Longwave Radiation Emitted by the Atmosphere

Longwave radiation emitted by the atmosphere was calculated through the Stefan–Boltzmann law, in which the emissivity of a body is proportional to the fourth power of the absolute temperature of said body. Thus,

$$L\downarrow = \varepsilon_a\,\sigma\,T^4 \tag{5}$$

where $\sigma = 5.67 \times 10^{-8}\,\mathrm{W\,m^{-2}K^{-4}}$ is the Stefan–Boltzmann constant, and $\varepsilon_a$ is the atmospheric emissivity, which was calculated through two empirical models proposed by Idso [36] (Equation (6)) and Prata [37] (Equation (7)), respectively.

The choice of these two models was based on previous analyses carried out in the Caatinga and Cerrado biomes [38–40]. Although these studies used different methods, they all agreed in indicating the models by Idso [36] and Prata [37] as appropriate in order to better represent atmospheric emissivity in the region. Specifically, a study [40] was carried out using in situ eddy covariance data measured in a preserved Caatinga segment. The same dataset was used in the calibration of the SITE model [31]. The quality of these data was previously analyzed and verified in other studies [21,24,30].

Idso's model [36] determines emissivity through observed air temperature data and estimated vapor pressure (Equation (6)). Similarly, Prata's model [37] determines emissivity based on air temperature and partial water vapor pressure (Equation (7)).

$$\varepsilon_{a,6} = 0.70 + 5.95 \times 10^{-7} \times e \times exp\left(\frac{1500}{T}\right) \tag{6}$$

$$\varepsilon_{a,7} = \{1 - (1 + \xi)exp[-(1.2 + 3.0\xi)^{0.5}]\}, with\ \xi = 0.465\left(\frac{e}{T}\right) \tag{7}$$

Although both models performed well in representing observed data, the model by Prata was slightly more advantageous regarding the mean absolute error [40]. Thus, we attributed weights to the emissivity computed by each model, with a 0.6 weight to Prata's model and a 0.4 weight to Idso's model. Finally, we calculated longwave radiation emitted by the atmosphere as follows:

$$L\downarrow = (0.4\varepsilon_{a.6} + 0.6\varepsilon_{a.7})\sigma\,T^4 \tag{8}$$

Since measured $L\downarrow$ data at the Ibimirim and Cabrobó sites are not available, the weighed emissivity solution (Equation (8)) was used in order to minimize errors inherent to the empirical nature of the formulations used for the calculation of the emissivity parameter.

### 2.4. Statistical Analysis

Linear trends were calculated by the non-parametric Mann–Kendall trend test [41,42]. This test consists of a comparison of each value of the time series with the remaining sequential values, calculating the number of times the remaining values are higher than the current tested value. The non-parametric Mann–Kendall trend test has been vastly used in trend analysis studies of meteorological variables in the NEB [43–45]. The following different statistical significance levels were used: 0.001, 0.01, 0.05, and 0.1 *p*-values. Trends tests were used for the entire air temperature, relative air humidity, and wind speed data series (January 2008 to December 2018). The results were analyzed as hourly trends and therefore are unprecedented to the region.

The statistical analyses were carried out in the R software [46], and figures were generated using the ggplot2 package [47]. Boxplots were used because they simultaneously represent different important statistical aspects of data, such as central tendency, variability, symmetry, and the presence of outliers. The boxplot shows three quartiles, and the minimum and maximum data in a vertically aligned rectangular box. The box comprises the interquartile interval, with its lower limit representing the first quartile (Q1) and its upper limit representing the third quartile (Q3). A dashed line within the box represents the second quartile (median). In both extremes of the box, there is a vertical line extending towards the maximum and minimum observed values. These are the so-called whiskers that extend to a distance of 1.5 from the limits of the box (Q3-Q1) (Chambers et al. (1983)). All data beyond these limits are highlighted as outliers (values lying too far from the majority), represented by blank circles. In addition, the mean value was also added to the boxplot represented by a filled circle.

## 3. Results

### 3.1. Seasonal Variability in Observed Meteorological Variables

Figure 2 shows the observed monthly rainfall variability in Cabrobó and Ibimirim in the years 2009 (wet year) and 2014 (dry year). In the wet year, Cabrobó registered higher rainfall volumes in all five initial months of the year, with the exception of May, when Ibimirim registered higher values. In the following months, both sites presented similar rainfall variability, with low precipitation volumes (0 to 50 mm) due to the establishment of the dry season. In December 2009, a rainfall peak (120 mm) can be observed in Ibimirim due to an early onset of the rainy season of the subsequent year (2010). In the dry year (2014), monthly rainfall values did not exceed 50 mm in Cabrobó. Similar behavior was observed in Ibimirim, with the exception of February, when more than 100 mm of precipitation was registered.

Figure 3 shows the air temperature in both studied sites and in the two contrasting years (2009 and 2014). In the wet season (first five months of the year), both sites present a similar air temperature behavior, but in the months from May to August, temperatures in Cabrobó were relatively higher, compared with Ibimirim, with the lowest values reaching 24 °C in Cabrobó and 22 °C in Ibimirim. Nevertheless, months with the highest and lowest temperatures coincided in both sites. In the dry period, the variability of air temperature was similar in both sites, but Cabrobó values exceeded 28 °C from September through December, while in Ibimirim, it only occurred in November.

In Cabrobó, the mean air temperature was higher in 2014 between February and June, than in 2009, which reflects the lower observed monthly precipitation. On the other hand, in Ibimirim, the mean air temperature was higher in 2014 during the months from April to July, confirming the influence of monthly rainfall distribution on temperature over the region. In the dry months (September to December), the temperature in 2014 presented a higher variability than in 2009, with temperature values lower than 21.5 °C (outliers) registered in Ibimirim, indicating that despite the lower annual accumulated rainfall that year, anomalous rainfall events can still drive the decrease in temperature.

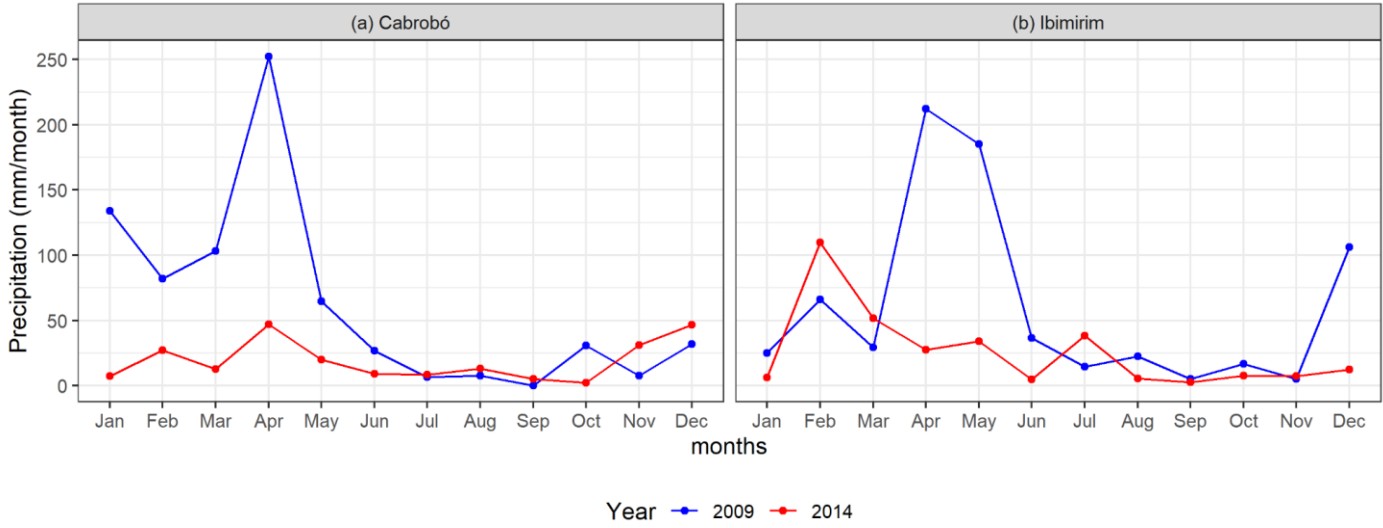

**Figure 2.** Observed monthly rainfall variability in Cabrobó and Ibimirim in 2009 (wet year) and 2014 (dry year).

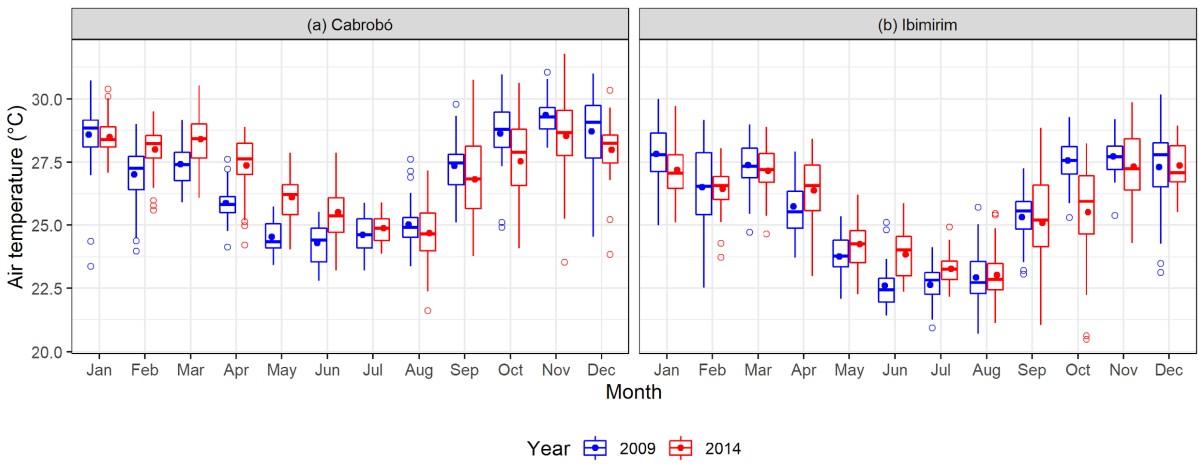

**Figure 3.** Monthly boxplots of hourly air temperature data in the Cabrobó and Ibimirim sites in 2009 (wet year) and 2014 (dry year).

Relative air humidity observed in Cabrobó and Ibimirim in 2009 and 2014 is shown in Figure 4. In the rainy season, both sites presented a similar monthly variability, with the highest RH value occurring in May. Additionally, higher RH values were observed during the five initial months of the years. In the dry season, Ibimirim presented higher RH values (greater than 80%), compared with Cabrobó, while also presenting a larger range in observed values.

Figure 5 shows the monthly wind speed variability in Cabrobó and Ibimirim, in 2009 and 2014. In the rainy season, wind speed varied from 1.0 to 5.0 m s⁻¹ in Cabrobó, while in Ibimirim, the range was smaller, with values between approximately 2.0 and 4.0 m s⁻¹. In the dry season, Cabrobó also presented higher wind speed values, with observations ranging from 2.0 to 5.0 m s⁻¹, while in Ibimirim, values ranged from 2.0 to 4.0 m s⁻¹.

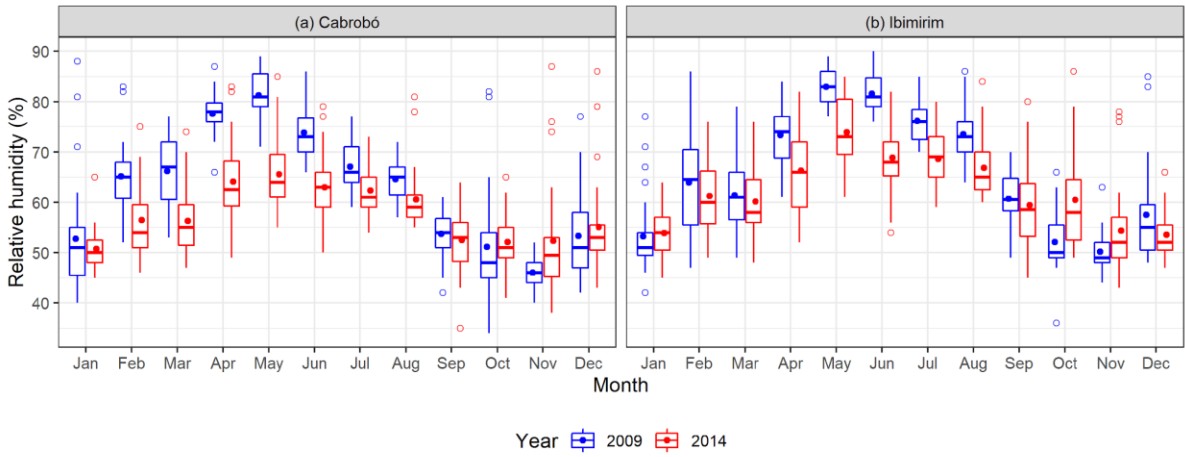

**Figure 4.** Monthly boxplots of hourly relative humidity data in the Cabrobó and Ibimirim sites in 2009 (wet year) and 2014 (dry year).

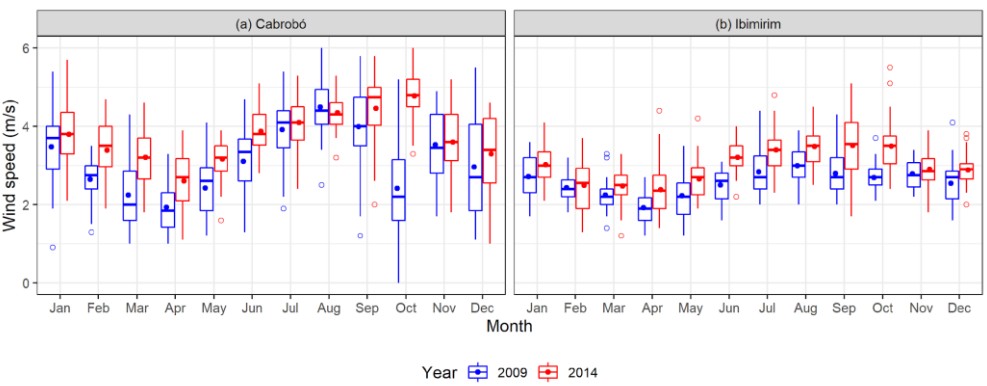

**Figure 5.** Monthly boxplots of hourly wind speed data in the Cabrobó and Ibimirim sites in 2009 (wet year) and 2014 (dry year).

*3.2. Hourly Linear Trends*

In Figure 6a,b, the results of the Mann–Kendall test for air temperature in the Cabrobó and Ibimirim sites, respectively, are shown. In Cabrobó, statistically significant trends were identified between 20:00 and 08:00, indicating increasing air temperatures during the night from December to August, which comprises the entire wet season in the region. On the other hand, no statistically significant trend was identified in the drier months (September to November). In Ibimirim, only a few sparsely distributed hours presented statistically significant trends. Despite the noticeable increase in temperature in Cabrobó (within the desertification hotspot), there were no significant temperature changes in Ibimirim (in the surroundings of the hotspot).

Trends results for relative humidity in both sites are shown in Figure 6c,d. In Ibimirim, increasing relative humidity trends were identified between August and October, mainly during night hours. In Cabrobó, however, an increasing trend was observed during the day from October to December, while sporadic negative trends were identified during the night from July to September.

In Figure 6e,f, the results of the Mann–Kendall test are shown regarding wind speed data in both sites. In Cabrobó, only March and April did not present significant trends. From May to June, significant decreasing wind speed trends were observed mostly from 9h to 11h. In the second half of the year (July to December), decreasing trends were identified during almost all hours of the day, especially from July to September, in which the results presented high statistical significance. In Ibimirim (Figure 6f), an opposite behavior was observed, compared with Cabrobó. The most significant trends were identified in

the six initial months of the year and indicated an increase in wind speed instead of a decrease. From August until the end of the year, significant trends were also identified mostly for the period from 12 h to 24 h.

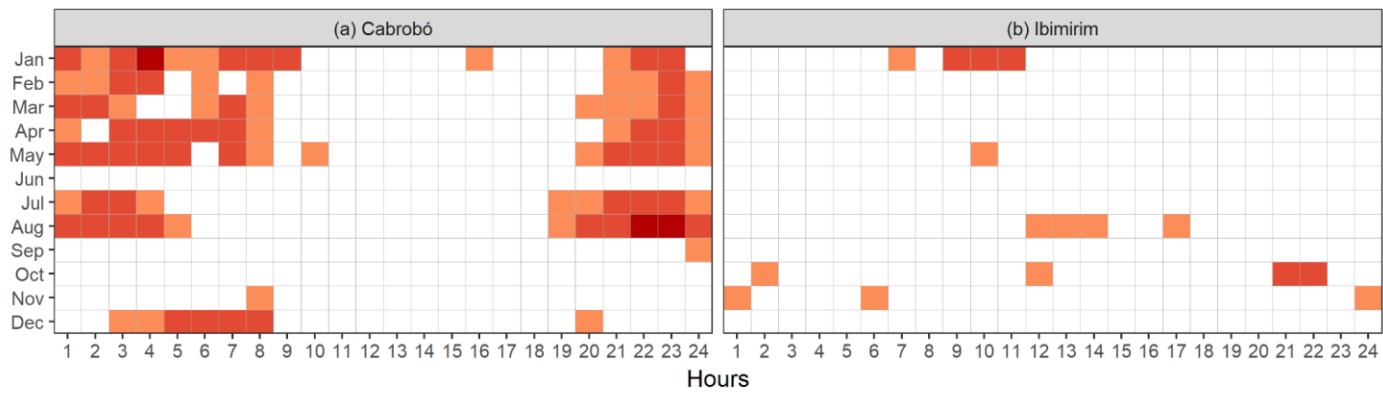

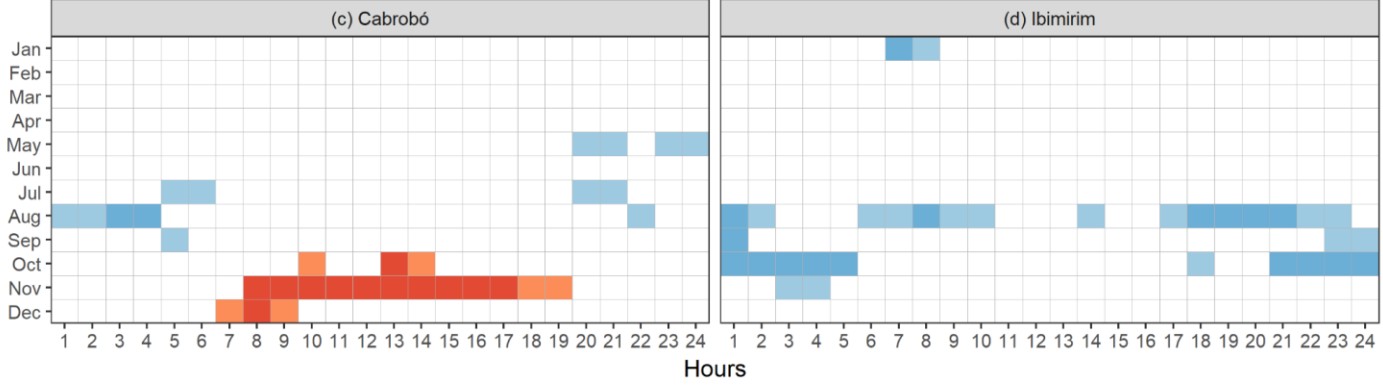

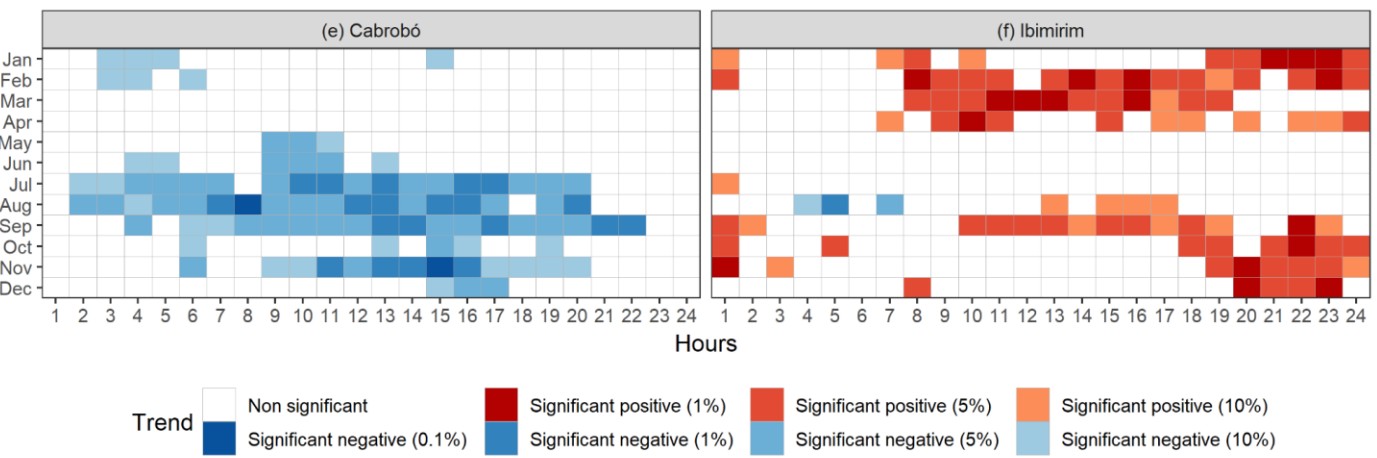

**Figure 6.** Hourly linear trends for air temperature (**a**,**b**), relative humidity (**c**,**d**), and wind speed (**e**,**f**). The first column refers to the Cabrobó site, and the second column refers to the Ibimirim site.

### 3.3. Simulated Components of Energy Balance

Figure 7 shows the monthly boxplots of simulated sensible heat flux (H) for the years 2009 and 2014 in each studied site. As it can be observed, the variability of H in the wet season is higher, ranging from 0 to 150 W m⁻². In the dry season, however, they ranged from negative values up to 100 W m⁻². Furthermore, both sites featured a similar monthly

distribution of H in both years, despite Cabrobó's series presenting a slightly more accentuated variability, with extreme values (outliers) registered throughout the year, particularly from October to December (both in 2009 and 2014).

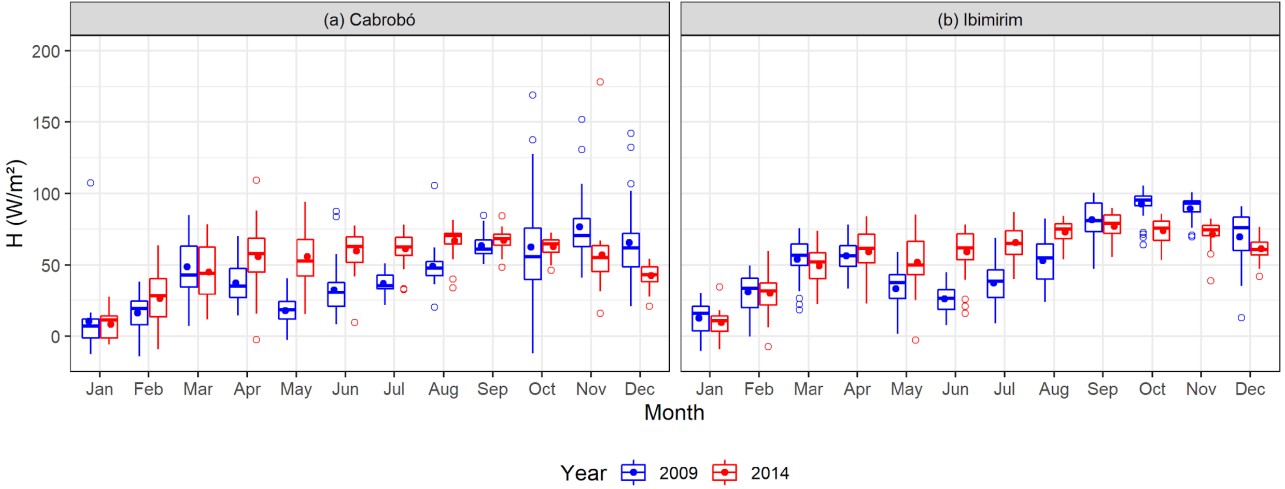

**Figure 7.** Monthly boxplots of hourly simulated sensible heat flux (H) in the Cabrobó and Ibimirim sites in 2009 (wet year) and 2014 (dry year).

The boxplots for the simulated latent heat flux (LE) in both sites for the years 2009 and 2014 are presented in Figure 8. Results show that the variability in LE in 2009 was higher than in 2014, in both sites. However, the simulated LE for the Cabrobó site presented a more remarkable variability, with values ranging from 0 to 450 W m$^{-2}$ in the dry season and from 0 to 400 W m$^{-2}$ in the wet season. Furthermore, the highest LE values were also found in Cabrobó, which reached up to 450 W m$^{-2}$. In both analyzed years, the values were higher during the initial three months of the year, which is expected due to the higher occurrence of rainfall these months. In 2009 and in 2014, in both sites, LE values decreased throughout the year with the establishment of the dry season, reaching the lowest values and variability in November and December.

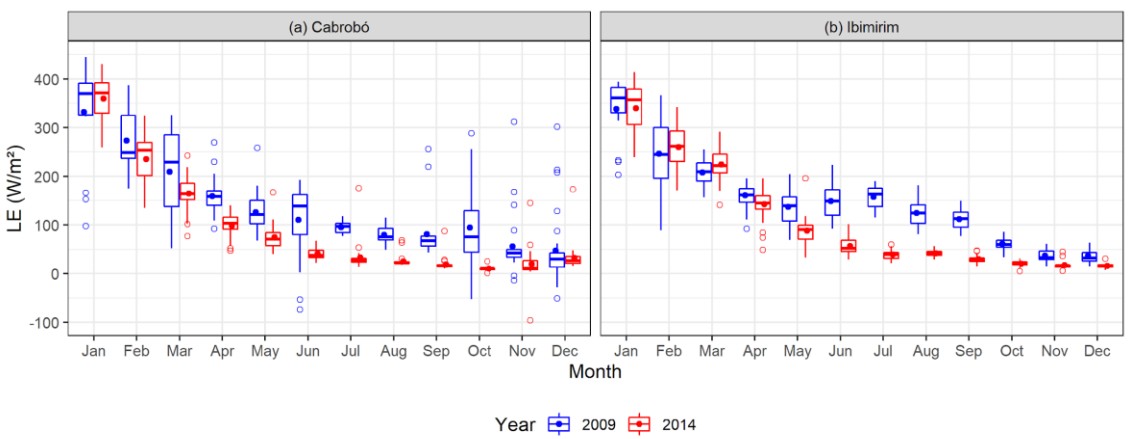

**Figure 8.** Monthly boxplots of hourly simulated latent heat flux (LE) in the Cabrobó and Ibimirim sites in 2009 (wet year) and 2014 (dry year).

Figure 9 shows the monthly boxplots of the Bowen ratio (ratio between H and LE) calculated for each site and year. In 2009, the Bowen ratio was small (near 0) in both sites during the initial months of the year, with no remarkable variability. In October, the Bowen ratio and its variability sharply rose, reaching up to 5.0. In 2014, however, there

was a gradual increase in the Bowen ratio throughout the year, which also peaked in the last three months of the year. In December, the ratio decreased roughly 60% in Cabrobó and 15% in Ibimirim.

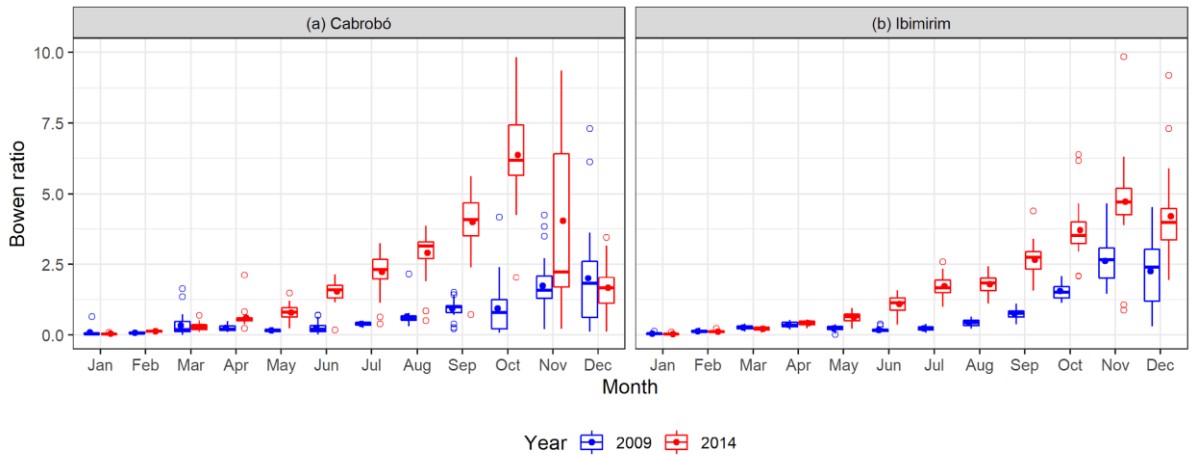

**Figure 9.** Monthly boxplots of hourly simulated Bowen ratio (ratio between sensible heat flux and latent heat flux) in the Cabrobó and Ibimirim sites in 2009 (wet year) and 2014 (dry year).

### 3.4. Simulated Components of CO₂ Balance

Boxplots of monthly values of gross primary production (GPP) are shown in Figure 10. In 2009, the simulated values ranged from 0.8 to 1.2 g C m⁻² h⁻¹ in the wet season and from 0.4 to 1.0 g C m⁻² h⁻¹ in the dry season, agreeing with the observed rainfall distribution. In 2014, GPP peaked in January in both sites, reaching values of approximately 1.2 g C m⁻² h⁻¹, followed by a gradual linear decrease in its values, reaching its lowest values (0.3 g C m⁻² h⁻¹) in December.

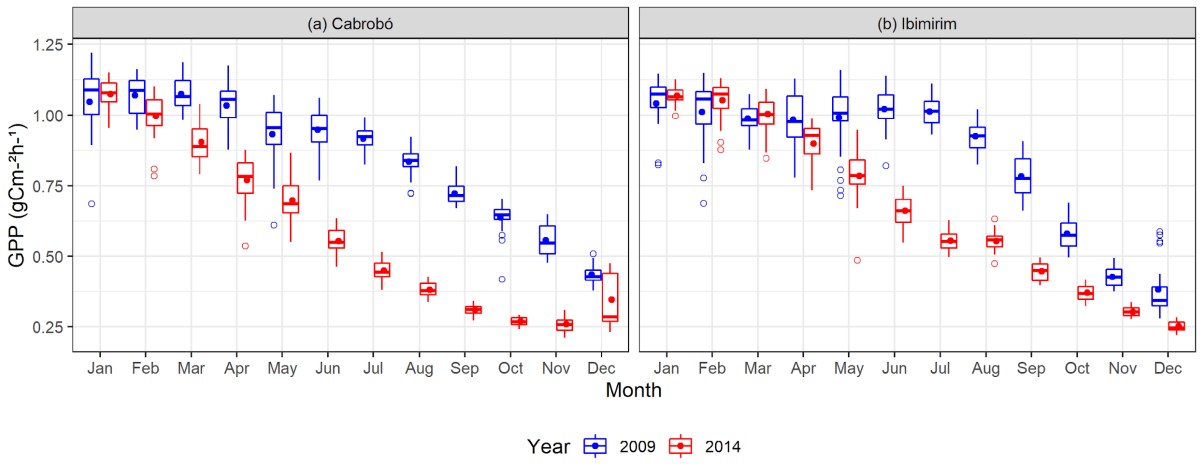

**Figure 10.** Monthly boxplots of hourly simulated gross primary production (GPP) in the Cabrobó and Ibimirim sites in 2009 (wet year) and 2014 (dry year).

Figure 11 shows the monthly boxplots of simulated net ecosystem exchange (NEE) for Cabrobó and Ibimirim in 2009 and 2014. One can observe that the behavior of simulated NEE is the opposite of simulated GPP, given that the more negative is the NEE, the higher is the carbon uptake. Lower NEE values (approximately −4 g C m⁻² h⁻¹) were found in the six initial months of 2009, when more carbon is absorbed by the ecosystem. From July onward, NEE values increased, indicating a reduction in CO₂ sequestration and a clear seasonal pattern. An increase in NEE values could also be observed in 2014, although

with a more linear characteristic, reaching up to 1 g C m$^{-2}$ h$^{-1}$ at the final trimester of the year, indicating that the release of $CO_2$ (ecosystem respiration) was higher than its absorption. In the wet season of 2009 (wet year), carbon uptake was higher in the first six months of the year, compared with 2014 (dry year).

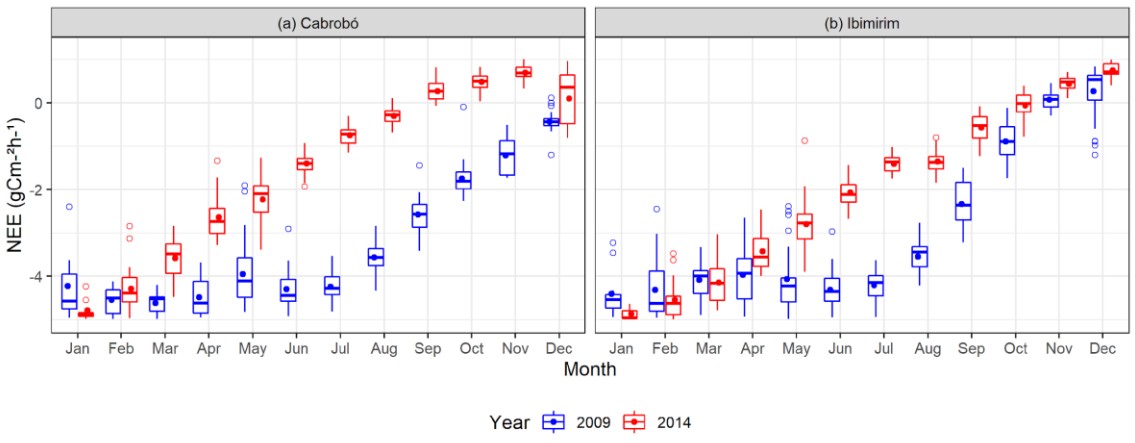

**Figure 11.** Monthly boxplots of hourly simulated net ecosystem exchange (NEE) in the Cabrobó and Ibimirim sites in 2009 (wet year) and 2014 (dry year). Regarding NEE values, negative values represent carbon uptake, while positive values represent carbon release.

## 4. Discussion

Comprehending the complex relationships between environmental factors and the components of energy and mass ($CO_2$) balances is crucial to combat the effects and expansion of desertification. In this study, the seasonal impact of the desertification process in meteorological variables measured in situ and in components of energy and $CO_2$ balances simulated by an SVAT model in NEB desertification hotspots was analyzed. Our results showed that despite the Cabrobó and Ibimirim sites being located within or near desertification hotspots under the influence of the same type of climate, meteorological variables behaved differently in each site. This is probably a consequence of the different partitioning of the energy balance at the surface.

In a region characterized by more degraded land (Cabrobó), sensible heat fluxes are expected to be higher than in less degraded lands (Ibimirim), while latent heat fluxes are expected to be lower. With a lower vegetation cover, the soil is increasingly exposed, increasing the risk of desertification [48,49]. Thus, with a higher transfer of sensible heat flux, the air near the surface becomes warmer. The simulations we carried out showed that there was a more remarkable variability (excess of H and LE outliers) in the Cabrobó site, which might indicate that it is indeed a more unstable region regarding energy balance, compared with the Ibimirim site.

Water availability conditions also play an important role in the development of vegetation, and our results showed that rainfall distribution was indeed a limiting factor for productivity in the studied sites, which is consistent with previous studies found in the literature [20,21,50,51]. In one study [21], mean NEE values in the wet season of a preserved Caatinga site was −0.70 g C m$^{-2}$ y$^{-1}$, while mean GPP was approximately 1.68 g C m$^{-2}$ y$^{-1}$. In the dry season, NEE values increased to approximately −0.26 g C m$^{-2}$ y$^{-1}$, while GPP values decreased to −0.64 g C m$^{-2}$ y$^{-1}$. Mendes et al. [21] also compared the net ecosystem $CO_2$ exchange (NEE) of the Caatinga biome with that of other savannas and tropical forests across the globe, showing that Caatinga could be considered mostly a carbon sink with mean assimilation of −157 g C m$^{-2}$ y$^{-1}$ in 2014 and 2015. The SITE model's simulated results obtained in this study are similar to those found by Mendes et al. [21], especially in terms of the Cabrobó site results.

Therefore, our results indicate that the SITE model can reasonably simulate seasonal variations in $CO_2$ fluxes in the studied regions. Results showed that even in the dry season, the NEE over the studied sites presented negative values (controlled by $CO_2$ assimilation), indicating that the Caatinga biome acted as a carbon sink in 2009 and 2014. The reduction in GPP and NEE (near 0) values with the decrease in precipitation is closely related to the reduction in leaf cover. In turn, this reduction is linked to leaf senescence of the Caatinga vegetation, which is a resilience mechanism in the face of drought that limits photosynthetic activity to the few semideciduous species that manage to keep their leaves all year round. During the dry season, trees are gradually impacted by the reduction in soil water content, which leads to the closure of stomates and to the reduction in stomatal conductance and leaf transpiration, which, in turn, limits $CO_2$ assimilation and reduces net photosynthesis.

The simulated values of the energy balance components agree with results from other studies carried out in Caatinga [24,52]. This provides further evidence that the SITE model is capable of simulating the partitioning of available energy into sensible and latent heat fluxes. For example, Campos [24] assessed the seasonal and annual behavior of energy partitioning and the energy balance closure in Caatinga. The authors identified a remarkable seasonal variability in the partitioning. Furthermore, they observed that during the dry period, more than half of the net radiation was converted into H, while in the wet season, this proportion reduced to less than 50%.

In the study carried out by Dos Santos et al. [52], the dry and wet seasons in the Caatinga region were analyzed. The authors observed that the conversion of energy into H is more expressive during the dry season. Mean LE values, on the other hand, reached approximately 200 W m$^{-2}$ during the wet season, while in the dry season, it was nearly zero. This behavior was also explained by the closure of stomates and leaf senescence in most Caatinga plants as an adaptation to drier conditions, reducing water loss through transpiration. Furthermore, the authors observed that H, instead of LE, is the main driver of energy balance in the biome during drought years, mostly due to deficits in soil water availability, which lead to low photosynthetic activity.

Other studies compared densely vegetated Caatinga sites with other sites with more sparsely and heterogeneous vegetation cover [53]. Results of these studies show that most parts of the net radiation were converted into H, as expected in semiarid environments. In these regions, LE surpassed H during the wet season, while the inverse occurred in the dry season. Furthermore, the difference in H and LE values during the dry season was more noticeable among studied sites. Additionally, LE was higher and more intense in the densely vegetated Caatinga region.

Low LE values in the Caatinga region may also represent a defense mechanism by plants, associated with leaf loss, which can be indirectly measured by remotely sensed vegetation indices such as the normalized difference vegetation index (NDVI). Deforestation affects the relationship between rainfall and NDVI [54], with degraded lands presenting lower productivity even in high water availability conditions. In preserved areas, however, NDVI responds more smoothly to rainfall variability, remaining relatively high even in drier conditions, compared with degraded lands.

## 5. Conclusions

We investigated seasonal and annual patterns for the components of energy and $CO_2$ balances during a wet (2009) and a dry (2014) year in a site (Cabrobó) known as a desertification hotspot in Northeast Brazil using biophysical soil–vegetation–atmosphere transfer and compared the observed trends with those recorded in a site (Ibimirim) in the same area but outside the hotspot. We found an increasing air temperature trend in the Cabrobó site, mainly during the night, which was not observed as remarkably in the Ibimirim site. Overall, there was an increasing air temperature trend in the Cabrobó site, mainly during the night, which was not observed as remarkably in the Ibimirim site. Regarding relative humidity, no trends were observed in the first semester. However, a few increasing trends

were observed in the Cabrobó site (particularly in November), while decreasing trends were observed in Ibimirim (mainly in August). Wind speed series presented an increasing trend in the Cabrobó site from June to August, while the opposite was observed in Ibimirim from September to May.

Sensible heat fluxes, latent heat fluxes, GPP, and NEE were satisfactorily simulated by the SITE model. GPP values were higher in the first half of the year in Ibimirim and Cabrobó, indicating that the $CO_2$ balance dynamics in Caatinga are closely linked to rainfall seasonality. NEE values were mostly negative throughout the year, indicating a higher $CO_2$ uptake even though minimum values were registered during dry months due to the lack of soil water. The results of this study show that the SITE model can be used to simulate adequate responses of energy and $CO_2$ fluxes for the Caatinga region, considering the remarkable phenological seasonality of this biome.

**Author Contributions:** Conceptualization, A.C.S., C.M.S.eS. and K.R.M.; methodology, A.C.S., C.M.S.eS. and K.R.M.; validation, A.C.S., C.M.S.eS., K.R.M., D.T.R. and P.R.M.; analyses formal A.C.S., C.M.S.eS., K.R.M., D.T.R., P.R.M., R.R.F. and B.G.B.; investigation A.C.S., C.M.S.eS. and K.R.M.; writing—original draft preparation, A.C.S., C.M.S.eS. and K.R.M.; writing—review and editing, A.C.S., C.M.S.eS., K.R.M., D.T.R., G.B.C., D.T.C.d.S., P.R.M., R.R.F. and B.G.B. All authors have read and agreed to the published version of the manuscript.

**Funding:** This research was funded by National Council for Scientific and Technological Development (CNPq), Process N°310781/2020-5 and National Observatory of Water and Carbon Dynamics in the Caatinga Biome (INCT-MCTI/CNPq/CAPES/FAPs 16/2014, grant: 465764/2014-2 and MCTI/CNPq N° 28/2018, grant 420854/2018-5).

**Institutional Review Board Statement:** Not applicable.

**Informed Consent Statement:** Not applicable.

**Data Availability Statement:** The data presented in this study are available on request from the corresponding author. The data are not publicly available due to privacy.

**Acknowledgments:** The authors are thankful to the Brazilian National Institute of Semi-Arid (INSA) for funding the project and providing the EC data used in this study. The authors are also thankful to the Coordination for the Improvement of Higher Education Personnel (CAPES) for the postdoctoral funding granted to the second author. This work was partially supported by the high-performance computing facilities of NPAD/UFRN. Finally, we are thankful to the National Council for Scientific and Technological Development (CNPq) for the research productivity grant of the C.M.S.eS. author (Process N° 310781/2020-5) and the financial support in the NOWCDCB project: National Observatory of Water and Carbon Dynamics in the Caatinga Biome (INCT-MCTI/CNPq/CAPES/FAPs 16/2014, Grant: 465764/2014-2 and MCTI/CNPq N° 28/2018, Grant 420854/2018-5).

**Conflicts of Interest:** The authors declare no conflict of interest.

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
