# Peer review of "Energy Balance, CO2 Balance, and Meteorological Aspects of Desertification Hotspots in Northeast Brazil"

_water, doi:10.3390/w13212962_

Round 1

Reviewer 1 Report

The paper is relevant for publication with minor changes. 

The author should improve the objectives by rephrasing them clearly. 

The theoretical framework should be added. 

Author Response

Response to Reviewer 1

  • The paper is relevant for publication with minor changes. 
  • Response: thanks!

  • The author should improve the objectives by rephrasing them clearly.
  • Response: ok, we change the text concerning the objectives.

  • The theoretical framework should be added.
  • Response: the theoretical framework has been revised in order to respond to the referees appointments in the new version of paper. 

Reviewer 2 Report

This manuscript presents an interesting case study. Although the results are limited to a comparison between only tow sites, they have relatively broad implications.

The research was well conducted, but presentation should be improved.

Manuscript organization

I think that lines 82-85 are incorrectly placed. They fit best at the end of the section (after line 99)

Perhaps, lines 138-140 are better placed immediately after “as illustrated in Figure 1.”

Cite the packages used for the analyses

Clarity

Line 26:  Gross primary production and net ecosystem exchange ->  Gross primary production (GPP) and net ecosystem exchange (NEE)

Lines 69-70: Please, explain briefly the meaning of “latent, sensible and soil heat fluxes” (I assume that plural is needed here). Probably, not many readers are familiar with these concepts.

Lines 89-93. This part presents the aim of the study and I suggest to present it in a slightly different, more incisive way: "the objectives of the present study were (1) to evaluate the role of the different components of the energy and CO2 balances in desertification hotspots in the semiarid NEB and (2) to assess the regional dynamics of the desertification process in relation to meteorological variables measured in situ".

Line 94: Explain here what the Bowen ratio is (it is explained only later in the ms)

Section 2.2. You cite in some parts 6 stations, in other 7. I realized that when you refer to the six stations, you refer to sites within the hotspot, whereas the seventh station is Ibimirim, which is outside the hotspot. Please, revise the text to make this clearer. Also, it is not clear what do you exactly mean with the % gaps (all types of measures?).

Lines 152-3: “correlation coefficients higher than 0.70 at a 95% confidence interval”. It is not clear what do you mean with “at a 95% confidence interval”. It seems that in that study, the authors calculated 95% CI for the r-values, which is fine, and found that their r-values were > than 0.7. Right?

Line 191: two empirical models (Eq. 6 and 191 Eq. 7) -> two empirical models proposed by Idso [39] (Eq. 6) and Parta [40] (Eq. 7), respectively.

Line 195: indicating the models [39-40] in order to better _> indicating Idso’s [39] and Parta’s [40] models as appropriate to

Line 200: The model [39] determines -> Idso’s model [39] determines

Line 201: Similarly, the model [40] determines -> Similarly, Parta’s model [40] determines

Lines 206-07: Were the weights subjective or defined following some rule?

Lines 212: Please, use the expression “non-parametric Mann-Kendall trend test” (trend is important to correct identify the test). I think that citing original papers by Mann and Kendall is not very useful. I would cite some more recent books or papers that explain the test. For example:

Wang, F., Shao, W., Yu, H., Kan, G., He, X., Zhang, D., Ren, M., & Wang, G. (2020). Re-evaluation of the Power of the Mann-Kendall Test for Detecting Monotonic Trends in Hydrometeorological Time Series. Frontiers in Earth Science.

Gilbert, R.O. 1987. Statistical methods for environmental pollution monitoring. Van Nostrand, New York.

Line 336: The monthly GPP boxplots -> Boxplots of monthly values of gross primary production (GPP)

Line 345: Figure 11 shows the monthly boxplots of simulated NEE -> Figure 11 shows the monthly boxplots of simulated net ecosystem exchange (NEE)

Language:

line 37: environmental issues -> environmental problems

lines 55 and 60: avoid repeating the expression “uttermost importance” (for example, use “are extremely important”)

line 71: due to land-use change, soil and atmosphere conditions -> due to land-use change, as well as soil and atmosphere conditions

Line 102: The meteorological data used was obtained through -> Meteorological data were obtained through

Line 108: They are measured -> They were measured

Line 149: similar to SITE, as reported in other study [29] -> similar to SITE [29].

Line 156: used in other study [31] -> used in another study [31]

Lines 182, 184, 190, etc.: use points, not commas for decimals

Lines 203 ff: Was reported [38] that despite both models performed well in representing observed data, the model by Prata was slightly more advantageous regarding the mean absolute error -> Despite both models performed well in representing observed data, the model by Prata was slightly more advantageous regarding the mean absolute error [38]

Line 250: I think spurious is not the best choice here. Perhaps, “anomalous”?

Lines 384-385: The sentence is grammatically incorrect. Perhaps you mean: “Mendes  et al. [21] also compared the net ecosystem CO2 384 exchange (NEE) of the Caatinga biome with that of other savannas and tropical forests across the globe, showing that the Caatinga could be considered mostly a carbon..”

Line 388: similar to those found [21], especially -> similar to those found by Mendes et al. [21], especially

Lines 405-6: The sentence is grammatically incorrect. Perhaps you mean:  For example, Campos [24] assessed the seasonal and annual behavior of energy partitioning and the energy balance closure in the Caatinga.

Line 406: You wrote “The authors identified …. Furthermore, they ….”. But if you are referring to reference 24, it is from a single author (Campos). Please check.

Line 410: In the study carried out by [47] -> In the study carried out by Dos Santos et al. [47]

Lines 434 and ff. They sound more like an Abstract than like a Conclusions. Please rephrase. Suggestion: “We investigated seasonal and annual patterns for the components of the energy and CO2 balances during a wet (2009) and a dry (2014) year a site (Cabrobó) a desertification hotspot in  Northeast Brazil using biophysical soil-vegetation-atmosphere transfer and compared the observed trends with those recorded in a site (Ibimirim) in the same area, but outside the hotspot. We found an increasing air temperature trend in the Cabrobó site, mainly  during the night, which was not observed as remarkably in the Ibimirim site.”

Other corrections

Figure 2. Months are not shown on the x-axis

Figures 7 and ff. Please, explain hat the boxplots are (median, interquartile range, min-max, outliers?)

Line 463: I cannot see how these data may have a privacy restriction. Please, make them public.

Author Response

Response to Reviewer 2

  • This manuscript presents an interesting case study. Although the results are limited to a comparison between only tow sites, they have relatively broad implications. 
  • Response: thanks!

  • The research was well conducted, but presentation should be improved.
  • Response: ok, we reviewed this.

Manuscript organization

  • I think that lines 82-85 are incorrectly placed. They fit best at the end of the section (after line 99) 
  • Response: Ok, modified.

  • Perhaps, lines 138-140 are better placed immediately after “as illustrated in Figure 1.”
  • Response: Ok, modified.

  • Cite the packages used for the analyses
  • Response: The statistical analyses were performed in the R language (Team, 2000), the figures were performed using the ggplot2 package (Wickham et al., 2016) of R language. (inserted in the text).

Clarity

  • Line 26:  Gross primary production and net ecosystem exchange ->  Gross primary production (GPP) and net ecosystem exchange (NEE)
  • Response: ok, entered.

  • Lines 69-70: Please, explain briefly the meaning of “latent, sensible and soil heat fluxes” (I assume that plural is needed here). Probably, not many readers are familiar with these concepts. 
  • Responde: ok, entered.

  • Lines 89-93. This part presents the aim of the study and I suggest to present it in a slightly different, more incisive way: "the objectives of the present study were (1) to evaluate the role of the different components of the energy and CO2 balances in desertification hotspots in the semiarid NEB and (2) to assess the regional dynamics of the desertification process in relation to meteorological variables measured in situ".
  • Responde: ok, modified.

  • Line 94: Explain here what the Bowen ratio is (it is explained only later in the ms)
  • Responde: ok, entered.

  • Section 2.2. You cite in some parts 6 stations, in other 7. I realized that when you refer to the six stations, you refer to sites within the hotspot, whereas the seventh station is Ibimirim, which is outside the hotspot. Please, revise the text to make this clearer. Also, it is not clear what do you exactly mean with the % gaps (all types of measures?).
  • Response: In fact, the description of stations is a little confusing, but we rewrote and corrected the sentences, clarifying the reviewer’s notes. 

  • Lines 152-3: “correlation coefficients higher than 0.70 at a 95% confidence interval”. It is not clear what do you mean with “at a 95% confidence interval”. It seems that in that study, the authors calculated 95% CI for the r-values, which is fine, and found that their r-values were > than 0.7. Right?
  • Response: In fact the correct term is “confidence level” and we corrected it in the text. The correlation coefficients higher than 0.70 were statistically significant at a 95% confidence level. 

  • Line 191: two empirical models (Eq. 6 and 191 Eq. 7) -> two empirical models proposed by Idso [39] (Eq. 6) and Parta [40] (Eq. 7), respectively.
  • Response: ok, modified.

  • Line 195: indicating the models [39-40] in order to better _> indicating Idso’s [39] and Parta’s [40] models as appropriate to
  • Responde: ok, modified.

  • Lines 206-07: Were the weights subjective or defined following some rule?
  • Response: as stated in the text we used as reference a local paper (reference 38) to give the weights, since the silver parameterization showed lower mean absolute error in this study we assigned a higher weight (0.6) to it in the weighted average.

  • Lines 212: Please, use the expression “non-parametric Mann-Kendall trend test” (trend is important to correct identify the test). I think that citing original papers by Mann and Kendall is not very useful. I would cite some more recent books or papers that explain the test. For example:
  • Response: ok, modified.

  • Wang, F., Shao, W., Yu, H., Kan, G., He, X., Zhang, D., Ren, M., & Wang, G. (2020). Re-evaluation of the Power of the Mann-Kendall Test for Detecting Monotonic Trends in Hydrometeorological Time Series. Frontiers in Earth Science.
  • Response: ok, modified.

  • Gilbert, R.O. 1987. Statistical methods for environmental pollution monitoring. Van Nostrand, New York.
  • Response: ok, modified.

  • Line 336: The monthly GPP boxplots -> Boxplots of monthly values o gross primary production (GPP)
  • Response: ok, modified.

  • Line 345: Figure 11 shows the monthly boxplots of simulated NEE -> Figure 11 shows the monthly boxplots of simulated net ecosystem exchange (NEE)
  • Response: ok, modified.

Language:

  • line 37: environmental issues -> environmental problems
  • lines 55 and 60: avoid repeating the expression “uttermost importance” (for example, use “are extremely important”)
  • line 71: due to land-use change, soil and atmosphere conditions -> due to land-use change, as well as soil and atmosphere conditions
  • Line 102: The meteorological data used was obtained through -> Meteorological data were obtained through
  • Line 108: They are measured -> They were measured
  • Line 149: similar to SITE, as reported in other study [29] -> similar to SITE [29].
  • Line 156: used in other study [31] -> used in another study [31]
  • Lines 182, 184, 190, etc.: use points, not commas for decimals
  • Lines 203 ff: Was reported [38] that despite both models performed well in representing observed data, the model by Prata was slightly more advantageous regarding the mean absolute error -> Despite both models performed well in representing observed data, the model by Prata was slightly more advantageous regarding the mean absolute error [38]
  • Line 250: I think spurious is not the best choice here. Perhaps, “anomalous”?
  • Lines 384-385: The sentence is grammatically incorrect. Perhaps you mean: “Mendes  et al. [21] also compared the net ecosystem CO2 384 exchange (NEE) of the Caatinga biome with that of other savannas and tropical forests across the globe, showing that the Caatinga could be considered mostly a carbon..”
  • Line 388: similar to those found [21], especially -> similar to those found by Mendes et al. [21], especially
  • Lines 405-6: The sentence is grammatically incorrect. Perhaps you mean:  For example, Campos [24] assessed the seasonal and annual behavior of energy partitioning and the energy balance closure in the Caatinga.
  • Line 406: You wrote “The authors identified …. Furthermore, they ….”. But if you are referring to reference 24, it is from a single author (Campos). Please check.
  • Line 410: In the study carried out by [47] -> In the study carried out by Dos Santos et al. [47]
  • Lines 434 and ff. They sound more like an Abstract than like a Conclusions. Please rephrase. Suggestion: “We investigated seasonal and annual patterns for the components of the energy and CO2 balances during a wet (2009) and a dry (2014) year a site (Cabrobó) a desertification hotspot in  Northeast Brazil using biophysical soil-vegetation-atmosphere transfer and compared the observed trends with those recorded in a site (Ibimirim) in the same area, but outside the hotspot. We found an increasing air temperature trend in the Cabrobó site, mainly  during the night, which was not observed as remarkably in the Ibimirim site.”
  • Response: all revisions of the language part have been modified.

Other corrections

  • Figure 2. Months are not shown on the x-axis 
  • Response: Months have been added on the x-axis in Figure 2.

  • Figures 7 and ff. Please, explain hat the boxplots are (median, interquartile range, min-max, outliers?
  • Response: we included in methodology a brief description of boxplots.

Reviewer 3 Report

formal aspects

the choice of the subscript for longwave radiation seems somewhat confusing to me. The downward arrow seems to indicate an incoming radiation, when in fact it is outgoing. Is it possible to put something else?

Throughout the text, the international and Latin notations for decimals are mixed. They should all be standardized in the international format (decimals separated by a period). Examples line 249 and equations 1, 2, 4, 5, 6, 7, 8 and line 190.

Figure 6 should indicate in the heading of each subfigure to which location they belong. As shown in figures 2,3,4, 5, 7, 8, 9, 10 and 11.

References should not contain the term "et al", instead they should list all the authors
e.g. references 5,6,7,8,9,9,10,11,11,15,20,21,28,28,29,45,46 and 48

aspects of the content
2.1 data. Authors should provide at least the average values of the most important climatic variables. A good place would be table 1, together with the position of each station.

The selection of the years 2009 and 2014 should be made based on statistical criteria and not their presumed anomaly. E.g. by specifying the probability of exceedance or similar parameter.

results. 
It seems clear that you did not measure any of the parameters provided by the model as an outcome. I remind the authors that the models must be properly adjusted and contrasted for their predictions to be minimally credible. Please explain this aspect. 

the conclusions only seem to highlight some of the results. it would be better if they provided some general and/or methodological ideas that other researchers could use as a reference for their own research.

Author Response

Response to Reviewer 3

  • the choice of the subscript for longwave radiation seems somewhat confusing to me. The downward arrow seems to indicate an incoming radiation, when in fact it is outgoing. Is it possible to put something else?
  • Response: Actually it is the atmospheric emission to the surface, so really the arrow is correct L↓.

  • Throughout the text, the international and Latin notations for decimals are mixed. They should all be standardized in the international format (decimals separated by a period). Examples line 249 and equations 1, 2, 4, 5, 6, 7, 8 and line 190.
  • Response: ok, modified. 

  • Figure 6 should indicate in the heading of each subfigure to which location they belong. As shown in figures 2,3,4, 5, 7, 8, 9, 10 and 11.
  • Response: The locations have been added in the heading of each subfigure in Figure 7.

  • References should not contain the term "et al", instead they should list all the authors
    e.g. references 5,6,7,8,9,9,10,11,11,15,20,21,28,28,29,45,46 and 48 aspects of the content
  • Response: ok, modified.

  • 2.1 data. Authors should provide at least the average values of the most important climatic variables. A good place would be table 1, together with the position of each station.
  • Response: we insert in a new table and add the comment about

  • The selection of the years 2009 and 2014 should be made based on statistical criteria and not their presumed anomaly. E.g. by specifying the probability of exceedance or similar parameter.
  • Response: we added a table (3) in the paper and included a justifying for the choice of 2009 and 2014.

Results

  • It seems clear that you did not measure any of the parameters provided by the model as an outcome. I remind the authors that the models must be properly adjusted and contrasted for their predictions to be minimally credible. Please explain this aspect. 
  • Response: the model was previously calibrated for the caatinga biome in a paper published by the group that composed the present research. That publication (31) showed that the model is able to adequately represent the variability of the components of the energy and CO2 balance in that biome when compared with data collected in situ through the Eddy Covariance technique. However, the objective of the present study was to use a model (already calibrated by Mendes et al., 2021) in a Caatinga environment that is in and around a desertification core (Cabrobó) (Ibimirim). It is important to draw attention to the fact that there are no data for the components of the carbon and energy balance in the desertification nuclei measured by the Eddy Covariance method. Therefore, it is necessary to clarify that the present research has this limitation regarding the comparison of simulations with measurements at the site of this study (Cabrobó and Ibimirim). On the other hand, the objective is to analyze the role of desertification in the components and, since there are no measurements, the model was used as a methodological tool and the results to be presented show that the model was able to capture the influence of the desertification process on the variables we propose to study. Finally, the input data for the model were collected in situ at the two sites analyzed.

  • the conclusions only seem to highlight some of the results. it would be better if they provided some general and/or methodological ideas that other researchers could use as a reference for their own research.
  • Response: we modified the conclusions and added information suggested by Referee  2.